# DET: Learn to Solve the Tunnel Traveling Salesman Problem using Double-Encoder Transformer

## Abstract

We delve into a challenging variant of the Traveling Salesman Problem (TSP), namely tunnel TSP, which incorporates a new important constraint requiring the traversal of a prescribed set of tunnels. While traditional deep reinforcement learning (DRL) based neural TSP algorithms excel in optimizing routes without tunnel restrictions, they often struggle to achieve optimal performance in tunnel TSP due to the neglect of the crucial role of tunnel attributes during solution generation. To address this challenge, we propose a simple but effective and flexible technique, called Double-Encoder Transformer (DET), which can be seamlessly integrated into various existing autoregressive neural TSP solvers. DET processes node and tunnel location information separately and encodes them in two distinct feature spaces. Following an efficient fusion strategy, DET then integrates the encoded information from nodes and tunnels, harnessing their intricate interactions. Experimental validation demonstrates that integrating DET into existing autoregressive neural solvers significantly improves performance, enabling us to reduce the average optimality gap for tunnel TSP from 12.58% (of the previous Single-Encoder model) to 7.35%.

## 1 Introduction

Combinatorial Optimization (CO) aims to find some optimal solutions within a combinatorial space, guided by an objective function and subject to constraints. The Traveling Salesman Problem (TSP) serves as a classic example, where the goal is to determine the shortest route that visits a set of fixed locations exactly once. However, real-world applications frequently involve additional complexities such as time constraints, capacity limitations, and infrastructure restrictions, necessitating the development of various TSP variants tailored to specific industries (Wahyuningsih & Sari, 2021; Toaza & Esztergár-Kiss, 2023).

One such variant is the Clustered TSP (CTSP) (Chisman, 1975). In CTSP, travelers are required to visit all nodes within a cluster before moving to other nodes. CTSP is an NP-hard problem (Mestria, 2018) and finds broad applicability in various domains, including route optimization in automated warehouses (Bock et al., 2024; Baniasadi et al., 2020), business transactions between supermarkets and suppliers, manufacturing (Laporte et al., 1998), integrated circuit testing, emergency vehicle dispatching and examination timetabling (Laporte & Palekar, 2002). Although CTSP has numerous applications and several existing algorithms specifically tailored to it (Mestria, 2018; Lu et al., 2020; Ahmed, 2014; Jiang et al., 2020; Bao et al., 2023; Dasari & Singh, 2023), most existing CTSP solvers employ heuristic methods, with no neural solvers specifically designed for this problem. Although (Chisman, 1975) proposes a method to reformulate a CTSP instance into a TSP instance, this approach is incompatible with current neural solvers since existing TSP neural solvers are built upon 2D Euclidean space, which cannot accommodate such a transformation. See Appendix A for a counterexample.

We focus on addressing the simplest special case of CTSP, where the cluster sizes are limited to a maximum of 2, and explore the potential of neural solvers for this problem. We refer to it as tunnel TSP, due to the clustering constraint resembling the traversal of a tunnel connecting two nodes. Although tunnel TSP may appear straightforward, it holds considerable significance. When priori-

tizing inter-cluster path optimization in CTSP, especially for large-scale TSP problems (Ding et al., 2007; Pan et al., 2023; Fu et al., 2021; Zong et al., 2022; Fu et al., 2023; Falkner & Schmidt-Thieme, 2023; He et al., 2023) or during CTSP algorithm testing, considering tunnel TSP can intuitively narrow down the solution space, enabling a quicker resolution. Initially, each cluster's interior is treated as a small-scale TSP problem, determining a path within it. Subsequently, we simplify the cluster's elements to two, representing the start and end points of the sub-TSP problem. This transformation converts the inter-cluster path optimization in CTSP into tunnel TSP, significantly enhancing the training efficiency for large-scale TSP problems.

In this paper, we systematically define tunnel TSP and introduce two Markov Decision Process (MDP) frameworks to model it. To enhance the utilization of tunnel TSP problem information, we propose a generalized strategy named DET (Double Encoder Transformer), which optimizes the solution process by separately inputting node information into a node encoder and tunnel information into a tunnel encoder to obtain their corresponding embeddings. During the decoding period, both encoder embeddings are concurrently considered for computation. While DET increases memory usage compared to a single encoder, it fully leverages tunnel information and explores its properties, yielding superior results.

In our experiments, we first compared the performance of several autoregressive TSP neural solvers for the tunnel TSP problem. Subsequently, we tested the performance of these algorithms after incorporating DET. The results show that DET, which can serve as a plugin seamlessly integrated into these DRL-based autoregressive TSP neural solvers, is compatible with both MDP procedures for solving tunnel TSP. Compared to those Single-Encoder algorithms, those incorporating DET significantly improve performance, enabling us to reduce the average optimality gap for tunnel TSP from 12.58% (using the Single-Encoder model) to 7.35%(using the Double-Encoder model). We firmly believe that this innovative approach has the potential to inspire further research into combinatorial optimization problems that involve specific spatial constraints. Our contribution can be concluded as follows:

- We address the tunnel TSP problem, a useful variant of TSP. To the best of our knowledge, no prior research has utilized deep reinforcement learning (DRL) to solve tunnel TSP problems.

- We proposed a model named DET, which encodes node and tunnel information distinctly to address tunnel TSP. DET can be seamlessly integrated as a plugin into various autoregressive neural TSP solvers. Experimental results show that incorporating the DET model enhances these solvers' ability to handle tunnel TSP problems effectively.

- By combining the proposed DET model with the Regret model, we develop the DET-POMO-Regret model. Experimental results showcase that this model achieves state-of-the-art (SOTA) performance on tunnel TSP problems of varying scales while preserving accuracy on the standard TSP problems.

## 2 RELATED WORK

**DRL neural methods for solving TSPs**   With the incorporation of DRL into the routing problems, some heuristics that are already commonly used achieve better results than the original heuristics (Ye et al., 2024; Zheng et al., 2023; d O Costa et al., 2020). Apart from these, starting with Pointer Network (Vinyals et al., 2015; Bello et al., 2016), more algorithms that leverage the characteristics of the problem are employed to solve the routing problem. Neural TSP solvers can be roughly divided into autoregressive solvers and Non-autoregressive solvers (Joshi et al., 2020). non-autoregressive solvers often apply GCN (Joshi et al., 2019; Fu et al., 2021) and GNN (Senuma et al., 2022) architectures, while autoregressive solvers often apply attention mechanism or recurrent neural networks. Attention Model (Kool et al., 2018) was the first autoregressive solver to use transformer (Vaswani, 2017). POMO (Kwon et al., 2020) exploits the symmetry of TSP routes and trains multiple instances simultaneously to enhance AM's efficiency. Furthermore, Pointerformer (Jin et al., 2023) employs a multi-pointer network to achieve better results. (Sun et al., 2024) introduces a regret mechanism that allows the autoregressive model to withdraw previous choices. Our DET is also based on autoregressive methods, which can be integrated into various autoregressive models.

Recent research on TSP problems has focused more on considering the problem structure to improve their generalization or reduce training costs. Sym-NCO (Kim et al., 2022b) proposes broader symmetries of different routing problems than POMO(Kwon et al., 2020). BQ-NCO (Drakulic et al., 2024) proposes a general framework to transform the CO problems into MDP. ELG (Gao et al., 2023) learns auxiliary strategies from locally transferable topological features, thereby improving cross-scale generalization performance. InViT (Fang et al., 2024) reduces the search space based on a statistical conclusion about the optimal solution. Starting from the unique structure of tunnel TSP, our DET greatly improves the generalization of different tunnel TSP problems by fully exploiting the structural characteristics of tunnel TSP.

**Different variants of TSPs**  To address complex real-world scenarios, researchers have developed many variations of the TSP problem. These variants include but are not limited to clustered TSP (Chisman, 1975), asymmetric TSP (Zhang et al., 2023), black and white TSP (Bourgeois et al., 2003), multi-commodity pickup and delivery TSP (Hernández-Pérez & Salazar-González, 2009; Ma et al., 2022) and generalized TSP (GTSP) (Pop et al., 2024). As heuristic algorithms have advanced, solvers like LKH3 (Helsgaun, 2017) have proven their proficiency in handling these TSP variants. However, the traditional heuristic algorithm faces the problem that the computation time increases greatly with the problem size.

The introduction of DRL has further broadened the horizon of TSP research. By harnessing the power of machine learning, researchers have explored new variants of TSP. (Zhang et al., 2021) introduces Dynamic TSP (DTSP), where the size of TSP instances is dynamically adjusted throughout the solution process. The most recently studied variant of the TSP problem is the min-max mTSP problem (Mahmoudinazlou & Kwon, 2024; Park et al., 2023; Kim et al., 2022a; Son et al., 2024; 2023), which seeks to minimize the maximum cost among all salesmen. These variants underscore the potential of TSP to address multiple optimization criteria. In addition, the DRL algorithm also greatly reduces the computation time (Kool et al., 2018; Jin et al., 2023), making it possible to solve large-scale problems. However, some pre-existing variants remain challenging to solve using DRL neural solvers, such as the CTSP and the GTSP. Our DET offers a solution to tunnel TSP, the simplest form of CTSP, marking the first attempt to use DRL to solve CTSP.

## 3 PROBLEM DEFINITION AND FORMALIZATION

In this section, we present the definition of the tunnel TSP problem, introduce some useful related concepts, and formalize its solution as a Markov Decision Process. We focus on the 2D Euclidean TSP and tunnel TSP problems.

### 3.1 TUNNEL TSP'S DEFINITION

**Traveling Salesman Problem**  A TSP instance with $m$ nodes can be described by an undirected graph $\mathcal{G} = (V, E)$, where $V = \{v_i \mid i = 1, \ldots, m\}$ represents the set of nodes and $E = \{e_{i,j} \mid i = 1, \ldots, m; j = 1, \ldots, m\}$ represents the set of edges. A feasible solution of a TSP instance can be defined as a closed cycle $\boldsymbol{\tau} = \{\tau_1, \tau_2, \cdots, \tau_m, \tau_1\}$, where $\tau_1, \tau_2, \cdots, \tau_m$ constitute a permutation of $v_1, v_2, \cdots, v_m$. The goal is to determine the optimal cycle $\boldsymbol{\tau}$ that minimizes the total cost $L_{TSP}(\tau)$, which can be defined as the sum of the costs incurred between consecutive nodes in the cycle $\boldsymbol{\tau}$. Formally:

$$L_{TSP}(\boldsymbol{\tau}) = \sum_{i=1}^{m} cost(\tau_i, \tau_{i+1}) \tag{1}$$

where $\tau_{m+1} = \tau_1$ for notational convenience.

**Tunnel TSP**  A tunnel TSP problem with $m$ nodes and $n$ tunnels can be described as a pair $(\mathcal{G}; S)$. Component $\mathcal{G} = (V, E)$ is the same as in a TSP instance, while component $S$ represents a set of tunnels, where a tunnel is simply a set of two node indices. Formally, $S = \{\{a_i, b_i\} \mid \forall i = 1, \ldots, n, a_i = 1, \ldots, m, b_i = 1, \ldots, m\}$. Without loss of generality, we assume that (1) a tunnel connects two different nodes and (2) no node belongs to two different tunnels. A feasible solution of a tunnel TSP instance can also be defined as a closed cycle $\boldsymbol{\tau} = \{\tau_1, \tau_2, \cdots, \tau_m, \tau_1\}$, In contrast to TSP, a feasible solution of tunnel TSP should ensure that for each tunnel $\{a_i, b_i\} \in S$, nodes $a_i$ and

$b_i$ are visited directly one after the other. This extra constraint guarantees that the tunnel's endpoints are directly connected in the solution sequence. The primary objective is to determine the optimal solution sequence $\boldsymbol{\tau}$ that minimizes the total cost $L_{tunnel}(\boldsymbol{\tau})$. This total cost is defined as the sum of the costs incurred between consecutive nodes in the cycle $\boldsymbol{\tau}$ minus a fixed distance component $D(S)$, which is a constant that represents the total length of the pre-determined tunnels specified in the set $S$.

$$L_{tunnel}(\boldsymbol{\tau}) = \sum_{i=1}^{m} cost(\tau_i, \tau_{i+1}) - D(S) \tag{2}$$

where $\tau_{m+1} = \tau_1$ for notational convenience. Given that both the total number of nodes and tunnels significantly impact the complexity of Tunnel TSP, it possesses two distinct variables that determine its scale, in contrast to TSP, which is defined by a single scale variable. For clarity, we adopt the notation **TTSP-m-n** to represent Tunnel TSP instances involving a total of $m$ nodes and $n$ tunnels. Also, we call a node 'connected node' if it is one end of an original tunnel. Otherwise, it is called a 'standalone node'.

## 3.2 Some Related Concepts

To facilitate our discussion, we introduce the following definitions:

**Definition 3.1.** *(Generalized Tunnel) A* generalized tunnel *is either an existing tunnel in the original problem or a dummy tunnel whose two endpoints correspond both to the same standalone node (not belonging to any existing tunnel).*

**Definition 3.2.** *(Corresponding Node) Node $v_i$ is called the* corresponding node *of node $v_j$ if they form the endpoints of some (possibly generalized) tunnel.*

Note that any node in $\mathcal{G}$ is either a standalone node or a connected node. Actually, $S$ is the set of original tunnels. We use $\bar{S}$ to represent the set of generalized tunnels. For a tunnel TSP instance $(\mathcal{G}; S)$, its $\bar{S}$ can be obtained easily from $\mathcal{G}$ and $S$. The number of generalized tunnels can be easily determined:

**Lemma 3.1.** *In a TTSP-m-n instance, there exist $m - n$ generalized tunnels.*

*Proof.* Each of the $n$ ordinary tunnels is directly counted. The remaining $m - 2n$ standalone nodes can be conceptually treated as generalized tunnels. □

## 3.3 The Markov Decision Process of Tunnel TSP

We can formalize the solution of a tunnel TSP instance using two different Markov Decision Processes (MDP). The first MDP starts from the definition of clustered TSP and restricts the agent's behavior based on a detector. Formally, the first MDP can be defined as follows:

**State** A state $s_t = (\mathbf{x}_t, \mathcal{G}, S)$, where $\mathbf{x}_t = (\tau_1, \tau_2, ..., \tau_t)$ represents the $t$-th partially complete feasible solution of tunnel TSP $(\mathcal{G}, S)$. $\tau_1$ can be any point in $V$.

**Action** The action is a node at time step $t$ that ensures the partial solution $\mathbf{x}_{t+1}$ is valid. i.e., $a_t \in V \backslash \{\tau_1, \cdots, \tau_t\}$. Due to the tunnel limitations, if $\tau_t$ is a connected node in $\mathcal{G}$ and its corresponding node has not been visited yet, then $a_t \in \{\hat{\tau}_t\}$, where $\hat{\tau}_t$ is the corresponding node of $\tau_t$.

**Transition** Given a state $s_t$ and an action $a_t$, the next state $s_{t+1} = (\mathbf{x}_{t+1}, \mathcal{G}, S)$ is deterministically determined with $\mathbf{x}_{t+1} = (\tau_1, \tau_2, ..., \tau_t, \tau_{t+1})$ with $\tau_{t+1} = a_t$.

**Reward** Every single time step has the reward $r_t = -cost(\tau_t, \tau_{t+1})$, which is defined as the negative cost of the corresponding chosen edge.

**Policy** Policy network $\boldsymbol{\pi_\theta}$ chooses actions according to the state $s_t$ at each time step $t$. For a given instance $(\mathcal{G}, S)$, the probability of a solution can be calculated as follows:

$$p_\theta(\mathbf{x}|\mathcal{G}, S) = \prod_{t=1}^{m} \pi_\theta(a_t|s_t) = \prod_{t=1}^{m} \pi_\theta(a_t|\mathbf{x}_t, \mathcal{G}, S) \tag{3}$$

where $m$ represents the total number of steps to get a feasible solution in this MDP because action is made on a per-node basis.

The second MDP utilizes the properties of tunnels. When a node is selected, the agent transitions directly to the corresponding node of the selected generalized tunnel. It seems more like the agent selects an endpoint of a generalized tunnel each time, taking the generalized tunnel as a unit. Formally, the second MDP can be defined as follows:

**State**  A state $s_t = (\mathbf{x}_t, \mathcal{G}, S)$, where $\mathbf{x}_t = (\tau_1, \tau_2, ..., \tau_{t'})$ represents the current partial feasible solution of tunnel TSP $(\mathcal{G}, S)$ at time step $t$, composed of a sequence of generalized tunnels. $\tau_1$ can be any point in $V$. For conciseness, we do not repeat the endpoints of a dummy tunnel. Therefore, $t' \geq t$.

**Action**  The action is a node at time step $t$ that ensures the partial solution $\mathbf{x}_{t+1}$ is valid, i.e., $a_t \in V \backslash \{\tau_1, ..., \tau_{t'}\}$.

**Transition**  After a state $s_t$ and an action $a_t$, the next state $s_{t+1} = (\mathbf{x}_{t+1}, \mathcal{G}, S)$ is determined deterministically either as $\mathbf{x}_{t+1} = (\tau_1, \tau_2, ..., \tau_{t'+1})$ with $\tau_{t'+1} = a_t$ if $a_t$ is a standalone node, or $\mathbf{x}_{t+1} = (\tau_1, \tau_2, ..., \tau_{t'+1}, \tau_{t'+2})$ with $\tau_{t'+1} = a_t$ and $\tau_{t'+2} = \hat{a}_t$ if $a_t$ is a connected node and $\hat{a}_t$ is its corresponding node.

**Reward**  Every single time step has the reward $r_t = -cost(\tau_{t'}, \tau_{t'+1})$, which is defined as the negative cost of the corresponding chosen edge.

**Policy**  Policy network $\boldsymbol{\pi_\theta}$ chooses an action at each time step $t$ given state $s_t$. For a given instance $(\mathcal{G}, S)$, the probability of a solution can be calculated as follows:

$$p_\theta(\mathbf{x}|\mathcal{G}, S) = \prod_{t=1}^{m-n} \pi_\theta(a_t|s_t) = \prod_{t=1}^{m-n} \pi_\theta(a_t|\mathbf{x}_t, \mathcal{G}, S) \tag{4}$$

where $m - n$ is the total number of steps to get a feasible solution in this MDP, by Lemma 3.1.

## 4 DOUBLE ENCODER TRANSFORMER

To solve tunnel TSP, we propose the Double Encoder Transformer (DET), which includes two encoders to process node and tunnel information in parallel. See Figure 1 for an overview of its pipeline. Recall that any node can be viewed not only as the node itself but also as an end of a generalized tunnel in the generalized tunnel graph.

**Input features of encoders**  We input the relevant features of nodes into the node encoder, formatted as $(x, y) \times 8$. Here, $(x, y)$ signifies the coordinate of a single node, the $\times 8$ indicates that we employed an 8-fold data augmentation strategy, a commonly used technique in previous work (Kwon et al., 2020), which generates 8 equivalent instances of each instance by flipping and rotating $\mathcal{G}$. We input the relevant features of generalized tunnels into the tunnel encoder, which has a form denoted as $(x_1, y_1, x_2, y_2; D_{tunnels}) \times 8$. Here $(x_1, y_1)$ and $(x_2, y_2)$ represent the coordinates of the two points that make up a generalized tunnel, $D_{tunnels}$ represents the enhancement aimed at emphasizing tunnel features, which may include multiple elements. Since directivity is naturally introduced in this tunnel representation, we use the symmetry method to avoid the bias in undirected tunnels. Specifically, for each tunnel, we create a 'mirror' version that has inverted coordinates and negative additional features (i.e. $(x_2, y_2, x_1, y_1, -D_{tunnels})$). The two versions are entered into the tunnel encoder together, and their encoding values are averaged to obtain the final tunnel encoding. This ensures that the encoding is direction-agnostic. In subsequent experiments, we set $D_{tunnels}$ to be the intrinsic length of the tunnel. For dummy generalized tunnels, $D_{tunnels}$ is set to 0.

**Numbers of different element inputs into the tunnel encoder**  We input the features of $m - n$ generalized tunnels into the tunnel encoder. This strategy aims to ensure that every node in $\mathcal{G}$ can be effectively integrated into the encoding process of the tunnel encoder. This strategy also guarantees that nodes located at both ends of the same generalized tunnel maintain consistency at the tunnel embedding level, meaning their corresponding tunnel embedding is identical.

**Generalized tunnel conversion table**  Following the encoding process, we introduce a transformation step using the generalized tunnel conversion table to convert the tunnel embedding into the

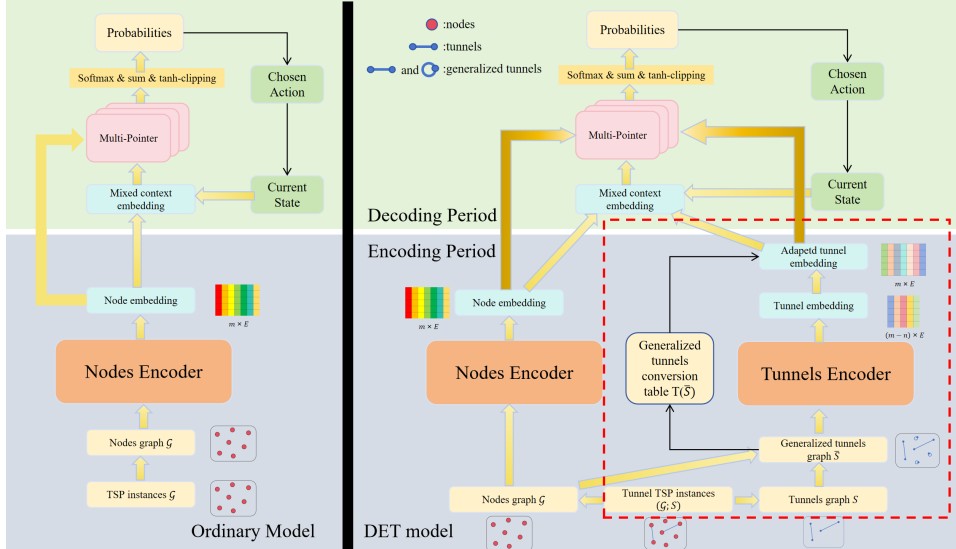

Figure 1: Pipeline of our DET. Our main difference is the introduction of a tunnel encoder, highlighted in the red dotted box. The DET model separates the tunnel TSP instance $(\mathcal{G}, S)$ into nodes graph $\mathcal{G}$ and tunnel graph $S$, then combines them into generalized tunnel graph $\bar{S}$. $\mathcal{G}$ is input into the node encoder to get the node embedding. The tunnel embedding is obtained by inputting $\bar{S}$ into the tunnel encoder, and the tunnel embedding is transformed into the same dimension as the node embedding through the generalized tunnel conversion table generated by the $\bar{S}$. In the decoding period, both embeddings are considered to calculate the probabilities.

same dimension as the node embedding. Specifically, the generalized tunnel conversion table $T(\bar{S})$ is defined as a transformation matrix that maps the tunnel embedding into a new dimension, enabling these embedding to directly correspond to the nodes in $\mathcal{G}$ while preserving the consistency of tunnel embedding for nodes located at both ends of the same generalized tunnel. In terms of formulation, $T(\bar{S})$ has the following form:

$$T(\bar{S})_{i,j} = \begin{cases} 1 & \text{if node } v_i \text{ belongs to the } j\text{-th tunnel in } \bar{S} \\ 0 & \text{Otherwise} \end{cases} \tag{5}$$

where $\bar{S}$ is the set of the generalized tunnels, containing $m - n$ elements as stated in Lemma 3.1.

**Decoding** In the decoding process, we need to consider both the embeddings of nodes and tunnels at the same time. DET can be easily integrated with various autoregressive neural solvers. The decoding process depends on the specific chosen model. We take the POMO (Kwon et al., 2020) algorithm as an example to elaborate a possible decoding scheme.

We employ the second MDP outlined in Section 3.3 to address this problem. We differentiate the information sources by using superscripts. Specifically, superscripts with '$o$' refer to information from node embedding, while superscripts with '$u$' refer to information from tunnel embedding. Firstly, similar to POMO, we obtain the current contextual embedding from the current state and both embeddings. Our query is constructed as follows:

$$\mathbf{q}^o = \mathbf{h}^{o,f} + \mathbf{h}^{o,l}, \mathbf{q}^u = \mathbf{h}^{u,f} + \mathbf{h}^{u,l} \tag{6}$$

where the superscript '$f$' signifies the fixed starting node of the partial route, while the superscript '$l$' indicates the dynamic ending node. Following this, we then utilize a multi-head attention mechanism. The required query has already been specified above, and both the keys and values originate from the respective embeddings:

$$\mathbf{A}^1 = \mathbf{MHA}(\mathbf{q}^o + \mathbf{q}^u, \mathbf{k}^o, \mathbf{v}^o), \quad \mathbf{A}^2 = \mathbf{MHA}(\mathbf{q}^o + \mathbf{q}^u, \mathbf{k}^u, \mathbf{v}^u) \tag{7}$$

Then we apply a Linear layer to project the $\mathbf{A}^1$ and $\mathbf{A}^2$ and compute the score using a final decoder layer with a single attention head. After that, we apply the $\tanh$ function to clip the score and mask

all visited nodes, including those automatically visited due to tunnel restrictions. The score for node $j$ is given by:

$$\boldsymbol{d}_j = \begin{cases} C \cdot \tanh(\frac{(W^{a1}\mathbf{A}^1 + W^{a2}\mathbf{A}^2)(\mathbf{k}_j^o + \mathbf{k}_j^u)}{\sqrt{d^k}}) & \text{if node } j \text{ is unvisited} \\ -\infty & \text{Otherwise} \end{cases} \tag{8}$$

where $W^{a1}, W^{a2}$ are both trainable Linear layers, $\boldsymbol{d}_j$ represents the score of node $j$. Finally, we compute the probability of node $j$ to be chosen using $softmax$ function:

$$\boldsymbol{p}_i = \pi_\theta(a_t = i | s_t, s) = \frac{e^{\boldsymbol{d}_i}}{\sum_j e^{\boldsymbol{d}_j}} \tag{9}$$

**Training**  Since the DET model does not require changes to the training strategy, we can use the same training method for DET-POMO as was used for POMO (Kwon et al., 2020). Specifically, we employ the REINFORCE algorithm (Williams, 1992) with rollout for training. We sample a set of $m$ trajectories $\{\boldsymbol{\tau}^1, \cdots, \boldsymbol{\tau}^m\}$, calculate the reward of each trajectory $f(\boldsymbol{\tau}^i)$. The gradient of the total training loss $\mathcal{L}$ can be approximated as follows:

$$\nabla_\theta \mathcal{L}(\theta) \approx \frac{1}{m} \sum_{i=1}^{m} [(f(\boldsymbol{\tau}^i) - b^i(s))\nabla \log p_\theta(\boldsymbol{\tau}^i | s)] \tag{10}$$

where $b^i(s)$ is a baseline function, which is commonly set as the average reward of those $m$ trajectories, serving as a shared baseline:

$$b^i(s) = b_{\text{shared}}(s) = \frac{1}{m} \sum_{i=1}^{m} f(\boldsymbol{\tau}^i) \text{ for all } i.$$

## 5 EXPERIMENTS

In this section, we begin by introducing our experimental settings. Following this, we present the key experimental results, highlighting the performance of various Single-Encoder models on the tunnel TSP problem and the enhancements observed after incorporating DET into each model. Lastly, we provide the results of ablation studies.

### 5.1 EXPERIMENT SETTING

To evaluate the efficiency of our DET, we compared the performance of the original Single-Encoder model and the model integrated with DET on tunnel TSP, using some state-of-the-art (SOTA) deep reinforcement learning (DRL)-based autoregressive TSP solvers as a baseline. All models were trained on a single Nvidia RTX 4090 24GB GPU using the same hyperparameters as in the original work. Training consisted of 500 epochs, with each epoch comprising 100 batches. The batch size varied depending on the problem size and available memory, ranging from 10 to 100 instances per epoch. For the TTSP-50-12 problem, each epoch took approximately 2.5 to 3 minutes to train.

**Datasets**  In our experiments, we use instances from TTSP_random to train various models corresponding to instances with different nodes. Although the TSPlib dataset is valid for the calculation of the TSP problem, it is inadequate for the tunnel TSP problem, which requires both TSP node information and tunnel connection information. To our knowledge, there is no dataset specifically designed for tunnel TSP.

- **TTSP_random**: For a TTSP-m-n problem, where $2n < m$, we uniformly sample $m$ nodes from the unit square $[0, 1]^2$. We then randomly select $2n$ numbers from the integer set $\{1, \cdots, m\}$, and pair them to form $n$ pairs, representing the ordinary tunnels. We generated 15 different scales of tunnel TSP problems based on $m = 50, 100, 200$ and $n = 0\%, 25\%, 50\%, 75\%, 100\%$. Here, $n = 25\%$ indicates that $25\%$ of the nodes are connected nodes. Due to significant memory consumption, we do not consider tunnel TSP problems with a larger number of nodes.

**Baselines** We consider the following DRL Algorithms as our baselines:

- **LKH3 (Helsgaun, 2017)**: LKH3 algorithm is an efficient heuristic algorithm for solving TSP, which uses local search and edge exchange strategies to find the optimal or near-optimal path solutions. To apply the LKH3 algorithm to the tunnel TSP problem, we follow the idea of (Lu et al., 2020) and set the elements corresponding to tunnels in the distance matrix to the negative values of the maximum distances between nodes, guiding the LKH3 algorithm to select tunnels actively.

- **POMO (Kwon et al., 2020)**: Leveraging the symmetry inherent in TSP problems, POMO utilizes N parallel instances to generate diverse trajectories during the training phase for dominance estimation. Additionally, it introduces a data augmentation technique to effectively reduce experimental variance.

- **InViT (Fang et al., 2024)**: InViT employs a nested design and incorporates invariant views within its encoders. This design enhances the model's generalization capabilities, enabling it to perform stably across problem instances with varying distributions or scales.

- **LCH-Regret (Sun et al., 2024)**: Building upon the existing learning construction methods, LCH-Regret introduces a learnable regret coding vector. This vector allows for rollback to the previous node during construction, avoiding the local optimal scheme caused by the greedy decoding strategy.

Table 1: Performance on different scales of tunnel TSP. The form of the element in the table is: Average Distance /GAP(%). The GAP % is w.r.t. the best value across all methods (usually LKH-3), and we omit the % sign. Bold refers to the best performance among all those DRL-based models.

| m & n | 50-0 | 50-6 | 50-12 | 50-18 | 50-25 |
|---|---|---|---|---|---|
| LKH3 | 5.692 | 4.935 | 4.152 | 3.379 | 2.476 |
| POMO | 5.731/0.67 | 5.686/15.21 | 5.271/26.94 | 4.646/37.51 | 3.333/34.60 |
| DET-POMO | 5.730/0.65 | 5.218/5.72 | 4.493/8.22 | 3.741/10.70 | 2.825/14.09 |
| InViT | 5.889/3.46 | 5.894/19.43 | 5.340/28.61 | 4.547/34.57 | 3.374/36.26 |
| DET-InViT | 5.888/3.44 | 5.766/16.84 | 5.015/20.78 | 4.248/25.72 | 3.382/36.59 |
| Regret | 5.717/0.44 | 5.255/6.48 | 4.647/11.91 | 3.852/13.99 | 2.670/7.85 |
| DET-Regret | **5.716/0.42** | **5.053/2.39** | **4.310/3.79** | **3.524/4.30** | **2.653/7.15** |

| m & n | 100-0 | 100-12 | 100-25 | 100-37 | 100-50 |
|---|---|---|---|---|---|
| LKH3 | 7.762 | 6.726 | 5.589 | 4.529 | 3.391 |
| POMO | 8.157/5.34 | 8.119/20.73 | 7.483/33.87 | 6.517/43.90 | 4.778/40.91 |
| DET-POMO | 7.985/3.11 | 7.455/10.85 | 6.505/16.37 | 5.492/21.26 | 4.340/27.99 |
| InViT | 8.197/5.85 | 8.201/21.94 | 7.361/31.69 | 6.308/39.28 | 4.673/37.81 |
| DET-InViT | 8.233/6.32 | 7.951/18.22 | 6.851/22.56 | 5.828/28.68 | 4.649/37.10 |
| Regret | 7.935/2.47 | 7.457/10.88 | 6.584/17.78 | 5.472/20.81 | 3.832/13.00 |
| DET-Regret | **7.858/1.48** | **7.111/5.74** | **5.992/7.20** | **4.954/9.38** | **3.791/11.78** |

| m & n | 200-0 | 200-25 | 200-50 | 200-75 | 200-100 |
|---|---|---|---|---|---|
| LKH3 | 10.703 | 9.216 | 7.759 | 6.279 | 4.800 |
| POMO | 11.736/9.65 | 11.806/28.09 | 10.797/39.14 | 9.335/48.67 | 6.858/42.87 |
| DET-POMO | 11.703/9.35 | 11.079/20.21 | 9.745/25.59 | 8.200/30.60 | 6.576/36.99 |
| InViT | 11.944/11.60 | 11.610/25.97 | 10.442/34.58 | 8.862/41.14 | 6.498/35.37 |
| DET-InViT | 11.966/11.81 | 11.225/21.80 | 9.770/25.92 | 8.327/32.62 | 6.478/34.96 |
| Regret | 11.459/7.07 | 10.797/17.15 | 9.610/23.86 | 7.990/27.24 | 5.652/17.75 |
| DET-Regret | **11.442/6.91** | **10.318/11.95** | **8.855/14.12** | **7.251/15.48** | **5.617/17.02** |

## 5.2 EXPERIMENT RESULTS

**Performance of different existing models** Considering the differences between tunnel TSP and the standard TSP, some methods that perform well in solving the standard TSP may not excel when addressing the tunnel TSP. Let's first consider the performance of different baseline methods.

We notice a significant widening of the performance gap when TSP models are applied to the tunnel TSP. Taking POMO as an example, the gap for solving TTSP-100-12 (20.73%) is nearly four times that of solving TTSP-100-0 (equivalent to TSP-100) at 5.34%. Similarly, POMO-Regret's gap for TSP-100-12 (10.88%) is also over four times that for TTSP-100-0 (2.47%). Notably, in some scenarios with a small number of tunnels $n$, we even observe that the results of some models numerically underperform their TSP-$m$ counterparts. As the number of tunnels increases, the performance gaps between models widen further. However, interestingly, when reaching TTSP-100-50, POMO's gap slightly narrows compared to TTSP-100-37. We believe that the reasons for this phenomenon are as follows:

Those connected nodes significantly elevate the problem's complexity and diminish the potential optimization space. Moreover, such tunnel constraints are more likely to lead algorithms to prematurely converge to suboptimal solutions within the search space. Additionally, TTSP-$m$-0 comprises solely of standalone nodes, TTSP-$m$-$\frac{m}{2}$ exclusively of connected nodes. In all other tunnel TSP instances, both standalone and connected nodes coexist. The Single Encoder model struggles to differentiate between these two types of nodes, thereby compromising its effectiveness. However, our Double-Encoder model distinguishes between these standalone nodes and connected nodes, contributing to our algorithm's superior performance.

**Compare DET-Model with Single-Encoder models** We have integrated the DET model into both the POMO and POMO-Regret models. It is noteworthy that the modifications in the Regret model are primarily made during the decoding period, where a regret flag is introduced during action selection for optimal use. In contrast, the alterations in DET centered around the encoding period and affected the method of calculating scores. Since these two models do not conflict, we can combine them and utilize the score calculation formula mentioned in Section 4 for decoding the DET-POMO-Regret framework. The results demonstrate that the DET-POMO model outperforms the POMO model across all problem scales, with particularly significant improvements observed when $n \neq 0, \frac{m}{2}$. In some cases, its performance is close to that of POMO-Regret. Thanks to the synergy between DET and Regret, DET-POMO-Regret further reduces the gap across all problem scales compared to both DET-POMO and POMO-Regret. DET-POMO-Regret represents the current best DRL solution for solving tunnel TSP. It is worth noting that the specific implementation details of the two encoders are not our primary concern, and we will later validate this in ablation studies.

We also adapted InViT to create DET-InViT, whose implementation details are shown in Appendix C due to space constraints. The InViT algorithm's effectiveness is partly attributed to the statistical prior that '98% of the path nodes in the optimal path of the TSP problem are within the 8-nearest neighbors of the corresponding node'. However, we found that this prior conclusion does not apply to the tunnel TSP problem, which may be the main reason why InViT's performance in the tunnel TSP problem is not much better than POMO and is significantly behind POMO-Regret. Regarding DET-InViT, although its performance is not significantly improved when $n = 0, \frac{m}{2}$, and may not be as good as InViT, it is still guaranteed to solve the tunnel TSP problem better than the original InViT when $n \neq 0, \frac{m}{2}$. But among the various DET models, DET-InViT is the one with the worst relative performance.

## 5.3 ABLATION STUDIES

In this part, we present some ablation experiment results that explain some important choices of our approach. All the experiments are conducted on the TTSP-50-18 problem in this part, using the DET-POMO-Regret model.

**The form of 2 encoders** We introduced two encoder frameworks called LinearNet (Vaswani, 2017) and RevNet (Gomez et al., 2017), and we will elaborate on their implementation details in the Appendix B. Our experimental results demonstrate that the specific type of framework employed by each of the two encoders has a negligible impact on the overall model performance, with a performance variation of less than 0.2%, as detailed in Table 2. This minimal fluctuation suggests that the detailed architectural design of the encoders does not significantly influence the final performance of the model, indicating a degree of robustness and insensitivity to architectural nuances within the encoder design. In all experiments in table 1, we used the LinearNet structure for the node encoder and the RevNet structure for the tunnel encoder.

Table 2: The effect of different Encoder structures and different numbers of elements fed into tunnel encoder.

| Node Encoder | Tunnel Encoder | Numbers of Elements | Result | Relative % |
|---|---|---|---|---|
| LinearNet | LinearNet | $m - n$ | 3.5277 | 100.04% |
| RevNet | LinearNet | $m - n$ | 3.5327 | 100.18% |
| LinearNet | RevNet | $m - n$ | 3.5263 | 100.00% |
| RevNet | RevNet | $m - n$ | 3.5317 | 100.15% |
| LinearNet | RevNet | $m$ | 3.7836 | 107.29% |

Table 3: The effect of different combinations when decoding. For brevity, columns 1 and 2 omit the MHA function, leaving only three parts of the input to MHA function. The function $cat(\cdot, \cdot)$ represents the concatenation behavior, which is the case when cross attention is considered. 'None' in column 2 indicates that there is no $\mathbf{A}^2$ in equation 8 calculation.

| $\mathbf{A}^1$ | $\mathbf{A}^2$ | 2nd Var in Equ. 8 | Result |
|---|---|---|---|
| $\mathbf{q}^o + \mathbf{q}^u, \mathbf{k}^o, \mathbf{v}^o$ | $\mathbf{q}^o + \mathbf{q}^u, \mathbf{k}^u, \mathbf{v}^u$ | $\mathbf{k}^o + \mathbf{k}^u$ | 3.5241 |
| $\mathbf{q}^o + \mathbf{q}^u, \mathbf{k}^o, \mathbf{v}^o$ | $\mathbf{q}^o + \mathbf{q}^u, \mathbf{k}^u, \mathbf{v}^u$ | $\mathbf{k}^o$ | 3.6080 |
| $\mathbf{q}^o + \mathbf{q}^u, \mathbf{k}^o, \mathbf{v}^o$ | $\mathbf{q}^o + \mathbf{q}^u, \mathbf{k}^u, \mathbf{v}^u$ | $\mathbf{k}^u$ | 5.9706 |
| $\mathbf{q}^o, \mathbf{k}^o, \mathbf{v}^o$ | None | $\mathbf{k}^o + \mathbf{k}^u$ | 3.5536 |
| $\mathbf{q}^u, \mathbf{k}^o, \mathbf{v}^o$ | $\mathbf{q}^u, \mathbf{k}^u, \mathbf{v}^u$ | $\mathbf{k}^o + \mathbf{k}^u$ | 5.6263 |
| $\mathbf{q}^o + \mathbf{q}^u, \mathbf{k}^o + \mathbf{k}^u, \mathbf{v}^o + \mathbf{v}^u$ | None | $\mathbf{k}^o + \mathbf{k}^u$ | 3.5324 |
| $\mathbf{q}^o + \mathbf{q}^u, cat(\mathbf{k}^o + \mathbf{k}^u), cat(\mathbf{v}^o + \mathbf{v}^u)$ | None | $\mathbf{k}^o + \mathbf{k}^u$ | 3.5380 |

**Numbers of different element inputs into the tunnel encoder**  Unlike the encoder architecture, the impact of different numbers of elements input into the tunnel encoder is quite significant. We considered the direct input of features for $n$ nodes within the tunnel encoder, finding that its performance was significantly inferior to that of input features for $m - n$ generalized tunnels. This highlights the importance of nodes situated at both ends of the same generalized tunnel sharing identical encodings in tunnel encoding. The outcomes of this comparison are presented in Table 2.

**Combination & cross-attention effects while decoding**  During the decoding stage, we proposed an approach that simultaneously considers two embeddings. However, this combination is not unique. We explored different combinations, and the detailed experimental results can be found in Table 3. Additionally, we considered cross-attention, where the dimensions of the query and key in the attention step are not aligned. The outcomes can be summarized as follows: omitting $\mathbf{q}^o$ in equation 7 significantly degrades performance, as does the absence of $\mathbf{k}^o$ or $\mathbf{k}^u$ in equation 8; completely disregarding tunnel information in equation 7 slightly degrades the result; while other variations, including the consideration of cross-attention, have minimal impact on the outcome. These results further justify the necessity of using two encoders.

## 6  CONCLUSION

In this work, we focus on an important variant of the TSP problem: tunnel TSP. It can be regarded as a special case of Clustered TSP. Notably, there is currently a lack of research on Clustered TSP using DRL. To fully exploit the information of tunnel TSP, we innovatively propose the DET architecture, which efficiently encodes node features and tunnel features separately and effectively integrates them. Through a large number of experimental validations, we demonstrate that DET can seamlessly integrate into various autoregressive neural TSP solvers, significantly enhancing their ability to solve tunnel TSP. We believe that the DET architecture offers new perspectives and approaches for solving combinatorial optimization problems with specific constraints and holds broad application prospects. In the future, we will continue to deepen this research by applying DRL methods to address more complex CTSP problems.

## Reproducibility

Our code is included in the supplementary materials, comprising the complete training code for DET-POMO, DET-InViT, and DET-POMO-Regret. You can easily adapt them into the original POMO, InViT, and POMO-Regret models. This facilitates the reproducibility of our work.

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

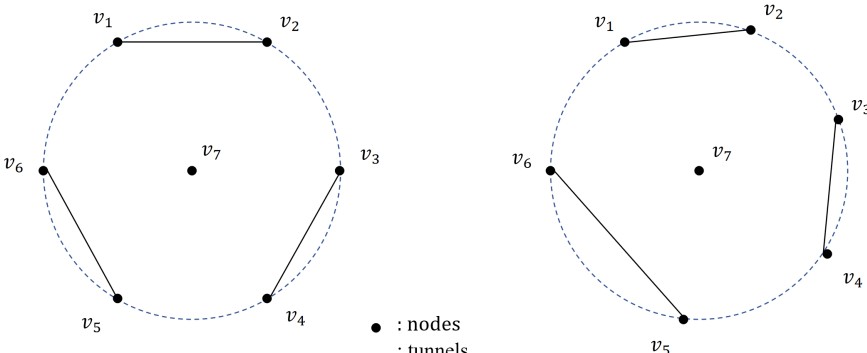

Figure 2: A simple example showing that CTSP transformations cannot be used in 2D Euclidean Spaces. On the left is the original Clustered TSP (CTSP) problem, and on the right is the transformed standard TSP problem. The contradiction arises at nodes $v_5, v_6$.

Jiongzhi Zheng, Kun He, Jianrong Zhou, Yan Jin, and Chu-Min Li. Reinforced lin–kernighan–helsgaun algorithms for the traveling salesman problems. *Knowledge-Based Systems*, 260: 110144, 2023.

Zefang Zong, Hansen Wang, Jingwei Wang, Meng Zheng, and Yong Li. Rbg: Hierarchically solving large-scale routing problems in logistic systems via reinforcement learning. In *Proceedings of the 28th ACM SIGKDD Conference on Knowledge Discovery and Data Mining*, pp. 4648–4658, 2022.

## A  THE TRANSFORMATION FROM CTSP TO TSP

It should be pointed out that the method of converting CTSP to TSP is carried out on the distance matrix. Given a TSP instance $\mathcal{G} = (V, E)$ with distance matrix $D$. Let $S = \{S_1, S_2, \cdots\}$ represent the set of clusters. Then the CTSP instance $(\mathcal{G}, S)$ can be transformed into TSP instance $\mathcal{G}' = (V', E')$ via the following steps:

- Define $V' = V$ and $E' = E$.

- Define the distance matrix $D'$ as:

$$d'_{ij} = \begin{cases} d_{ij} + C & \text{if } v_i \text{ and } v_j \text{ belong to different clusters} \\ d_{ij} & \text{Otherwise} \end{cases} \tag{11}$$

where $C$ is a sufficiently large constant. The above operation is done on the distance matrix. Let's assume the following problem as shown in the left part of Figure 2. There are 7 points and 3 tunnels (clusters), and the original problem is defined in 2D Euclidean space. Points $v_1, v_2, \cdots, v_6$ are evenly distributed on a unit circle centered around point $v_7$. Then its distance matrix is:

$$\begin{bmatrix} 0 & 1 & \sqrt{3} & 2 & \sqrt{3} & 1 & 1 \\ 1 & 0 & 1 & \sqrt{3} & 2 & \sqrt{3} & 1 \\ \sqrt{3} & 1 & 0 & 1 & \sqrt{3} & 2 & 1 \\ 2 & \sqrt{3} & 1 & 0 & 1 & \sqrt{3} & 1 \\ \sqrt{3} & 2 & \sqrt{3} & 1 & 0 & 1 & 1 \\ 1 & \sqrt{3} & 2 & \sqrt{3} & 1 & 0 & 1 \\ 1 & 1 & 1 & 1 & 1 & 1 & 0 \end{bmatrix}$$

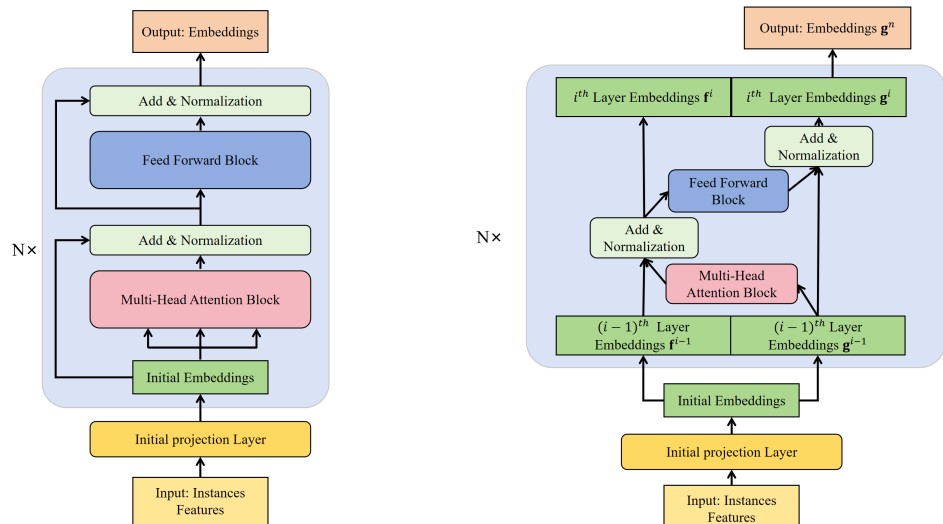

Figure 3: Specific structure of 2 different Encoders. LinearNet on the left and RevNet on the right.

According to the Equation 11, the transferred distance matrix is:

$$\begin{bmatrix} 0 & 1 & \sqrt{3}+C & 2+C & \sqrt{3}+C & 1+C & 1+C \\ 1 & 0 & 1+C & \sqrt{3}+C & 2+C & \sqrt{3}+C & 1+C \\ \sqrt{3}+C & 1+C & 0 & 1 & \sqrt{3}+C & 2+C & 1+C \\ 2+C & \sqrt{3}+C & 1 & 0 & 1+C & \sqrt{3}+C & 1+C \\ \sqrt{3}+C & 2+C & \sqrt{3}+C & 1+C & 0 & 1 & 1+C \\ 1+C & \sqrt{3}+C & 2+C & \sqrt{3}+C & 1 & 0 & 1+C \\ 1+C & 1+C & 1+C & 1+C & 1+C & 1+C & 0 \end{bmatrix}$$

where $C \neq 0$. When considering the new distance matrix as representing 7 points in a 2D Euclidean space, it becomes apparent that points $v_1, v_2, \cdots, v_6$ are distributed on the circle with point $v_7$ as the center and $1 + C$ as the radius. We can easily observe that $\angle v_6 v_7 v_1 = \angle v_2 v_7 v_3 = \angle v_4 v_7 v_5 = 60°$. However, since $C \neq 0$, we have $\angle v_1 v_7 v_2 \neq 60°, \angle v_3 v_7 v_4 \neq 60°, \angle v_5 v_7 v_6 \neq 60°$, thus the sum of all angles around the circle does not equal to $360°$, leading to a contradiction. This counterexample shows that this method of converting CTSP (tunnel TSP) to TSP cannot be implemented in 2-dimensional Euclidean space. This shows that we can not directly use the neural solver to get the exact solution of tunnel TSP.

## B    THE SPECIFIC ENCODER STRUCTURES

In this section, we will show two different encoder structures, both of which can be used in the encoder module in Figure 1.

### B.1    LINEARNET

The structure of this encoder can be seen in the left-hand section of Figure 3. Its structure is similar to the encoding structure in Transformer (Vaswani, 2017). First, we pass the input through an initial mapping layer to obtain the initial mapping $\mathbf{h}^1$. Following this, a series of sequential Multi-Head Attention (MHA) and Feed-Forward (FF) processing steps are performed, where after each processing step, a simple addition is carried out, followed by layer normalization. This results in the final embedding output. Mathematically, this series of processing steps can be represented as follows:

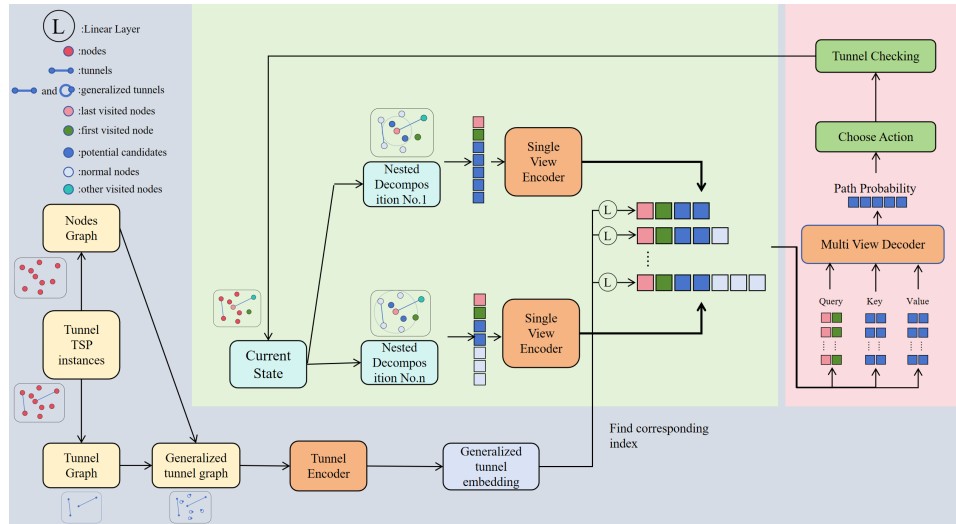

Figure 4: The pipeline of DET-InViT. The main adaptation occurs in the gray area.

$$\hat{\mathbf{h}}_i = \mathbf{LayerNorm}^{(l)}(\mathbf{h}_i + \mathbf{MHA}^{(l-1)}(\mathbf{h}_1^{(l-1)}, \cdots, \mathbf{h}_n^{(l-1)}))$$
$$\mathbf{h}_i^{(l)} = \mathbf{LayerNorm}^{(l)}(\hat{\mathbf{h}}_i + \mathbf{FF}^{(l-1)}(\hat{\mathbf{h}}_i)) \qquad i = 1, \cdots, N \tag{12}$$

Here 'MHA' means the trainable multi-head attention layer, and 'FF' means the trainable feed-forward layer. $l = 1, \cdots, N$ indicates that different layers do not share parameters, while $n$ means the number of heads. In our experiment, $N = 6, n = 8$.

## B.2 REVNET

RevNet, introduced in (Gomez et al., 2017), incorporates a unique mechanism that facilitates the direct calculation of derivatives during the backpropagation process, thereby minimizing the memory footprint of encoding.. the structure of this encoder can be seen in the right-hand section of Figure 3. In RevNet, a pair of input vectors $(\mathbf{f}^1, \mathbf{g}^1)$ are processed through an alternating sequence of Multi-Head Attention (MHA) and Feed-Forward (FF) modules, enabling the computation of the subsequent pair $(\mathbf{f}^{i+1}, \mathbf{g}^{i+1})$. To initialize the network, we first obtain $\mathbf{h}^1$ by projecting the input features through an initial layer. Then we duplicate $\mathbf{h}^1$ and feed the resultant pair $(\mathbf{h}^1, \mathbf{h}^1)$ as the initial input $(\mathbf{f}^1, \mathbf{g}^1)$ to RevNet. Subsequently, this input is passed through several reversible blocks, each comprising MHA and FF modules in a reversible manner. This process culminates in the generation of the final pair $(\mathbf{f}^o, \mathbf{g}^o)$. The process can be expressed in the following formulas:

$$\mathbf{f}^1 = \mathbf{g}^1 = \mathbf{h}^1,$$
$$\mathbf{f}^{i+1} = \mathbf{f}^i + \mathbf{MHA}^i(\mathbf{g}^i). \ \ i = 1, \cdots, N.$$
$$\mathbf{g}^{i+1} = \mathbf{g}^i + \mathbf{FF}^i(\mathbf{f}^i). \ \ i = 1, \cdots, N. \tag{13}$$

In our experiment $N = 6$. 'MHA' means the trainable multi-head attention layer, and 'FF' means the trainable feed-forward layer. The superscript of MHA and FF indicates that different layers do not share parameters. Actually, the second line and third line of the formula 13 is how RevNet works. Ultimately, $\mathbf{g}^{n+1}$ is output as the final embedding.

## C  ADAPTING INVIT MODEL INTO DET-INVIT

Due to InViT's unique encoding mechanism (Fang et al., 2024), although it is a DRL-based autoregressive TSP solver, its Double-Encoder implementation diverges from POMO and POMO-Regret.

Instead, InViT extracts the current node and several of its closest unvisited neighbors, encodes them, and computes attention. In this part, we'll cover the conversion of the double-encoder to the InViT model in detail. It is noteworthy that in the DET-InViT model, we adopt the first MDP in section 3.3, which operates by restricting the agents' actions based on a detector.

**Global tunnel encoder**    Our primary modification involves the introduction of a global tunnel encoder that employs the principle of the tunnel encoder discussed in Section 4. This encoder is tasked with encoding $m - n$ generalized tunnels accommodating both positive and negative input directions. Ideally, we would encode one tunnel encoder for each state encoder and one for each action encoder. However, considering the storage cost, we opted to use a single global tunnel encoder, When integrating tunnel information into each state or action encoder, we prepend an independent linear layer to facilitate the conversion process, and the generalized tunnel transformation matrix is utilized after this step. Since DET does not alter the training methodology, we train DET-InViT in the same manner as InViT.

**Combination strategy**    Put simply, in InViT, the embeddings of the last visited node and the first visited node (or depot) serve as the query input for the decoder, while the embeddings of other potential candidates act as the keys and values within the decoder. After encoding each generalized tunnel, its encoded value is added to the state encoder and action encoder at each step through numerical summation. We offer two distinct fusion methods: add tunnel embedding only to the query in the decoder; and add tunnel embedding to all the queries, keys, and values in the decoder. As stated in Section 5.2, the statistically derived prior conclusion that '98% of the path nodes in the optimal path of the TSP problem are within the 8-nearest neighbors of the corresponding node' is the basis of the outperformance of InViT, but it may not fit for tunnel TSP. When the number of both standalone nodes and connected nodes is not very small, adding tunnel information to the key and value can greatly improve the results, as it allows for better distinction between the two types of nodes. Therefore, when $n$ is large ($n > 0.45m$, i.e. 90% of the nodes are connected nodes), and the number of standalone nodes is relatively small, we only add tunnel information to the query from InViT's Multi-view Decoder; Similarly, when the number of $n$ is small ($n < 0.05m$), the number of connected nodes is relatively small, and we only add tunnel information to the query. Otherwise, we add tunnel information to all the queries, keys, and values in the decoder. In mathematical terms, if we denote the multi-view decoding process as a function like $\mathbf{MVD}(\mathbf{q}, \mathbf{k}, \mathbf{v})$, then the original decoding function of InViT can be expressed as:

$$\mathbf{d} = \mathbf{MVD}(\mathbf{q}^o, \mathbf{k}^o, \mathbf{v}^o)$$

where $\mathbf{d}$ is the score of different nodes. The superscript 'o' represents information from node embedding which is generated by the original InViT. Then the decoding function of our DET-InViT can be expressed as:

$$\mathbf{d} = \begin{cases} \mathbf{MVD}(\mathbf{q}^o + \mathbf{q}^u, \mathbf{k}^o + \mathbf{k}^u, \mathbf{v}^o + \mathbf{v}^u) & \text{if } 0.05m \leq n \leq 0.45m \\ \mathbf{MVD}(\mathbf{q}^o + \mathbf{q}^u, \mathbf{k}^o, \mathbf{v}^o) & \text{if } n < 0.05m \text{ or } n > 0.45m \end{cases}$$

where $m, n$ shares the same meaning as TTSP-m-n. The superscript 'u' represents information from tunnel embedding which is generated by the new-added global tunnel encoder.

**Tunnel checking**    When adopting a multi-instance synchronous training strategy similar to POMO (Kwon et al., 2020), we streamline the generation process of the action space, temporarily overlooking the restrictions imposed by tunnels on the action space. However, to adhere to the constraints of the Tunnel TSP problem, a dedicated tunnel checking step is necessary after decoding and calculating the corresponding actions and probabilities. The primary purpose of tunnel checking is to verify whether the selected actions comply with the constraints of the Tunnel TSP problem and to rectify them if they do not. Specifically, this step checks whether the last visited node is a connected node in graph $\mathcal{G}$ and whether its corresponding node has not been visited yet. If these conditions are not met, the original action and probability are retained; otherwise, the action is adjusted to visit its corresponding node, and its selection probability is set to 1.

