# OpenReview forum: "DET: Learn to Solve the Tunnel Traveling Salesmen Problem using Double-Encoder Transformer"
_ICLR.cc/2025/Conference — ICLR 2025 Conference Withdrawn Submission_

### Official Review · Reviewer_XdTB · 2024-11-03

**Soundness:** 3
**Presentation:** 3
**Contribution:** 2
**Rating:** 5
**Confidence:** 3

**Summary:**

This paper proposes a challenging variant of Clustered Traveling Salesman Problem (CTSP), called tunnel TSP, which incorporates an important constraint requiring the traversal of a prescribed set of tunnels. The authors utilize deep reinforcement learning (DRL) for this problem, where the method is called Double Encoder Transformer (DET). It encodes node and tunnel information and can be applied to the existing method to solve tunnel TSP problems. The experimental results show the effectiveness of the DET model on various scaled problems.

**Strengths:**

- It clearly redefines tunnel TSP task with a notation TTSP-m-n, where there are a total of m nodes and n tunnels. In this setting, a node can be connected or standalone, and the cost is similar to the original version, but includes a fixed distance, D(S).
- The model utilizes two separate transformer encoders, which encode different information like node and tunnel. It enhances the overall performance via distinctly encoding tunnel information from graphs.
- The proposed method can effectively solve scale-variant tunnel TSP problems, which is hard for existing approaches.

**Weaknesses:**

- It is unclear why the tunnel TSP task is important to systematically define and resolve. This task looks like a simple variation of CTSP. Please provide real-world examples to support its importance.
- The explanation lacks clarity on why two encoders are necessary and what specific motivation supports this design choice. In addition, the overall method seems really simple and lacks a strong sense of novelty.
- While this may be the first application of DRL to CTSP, its novelty is questionable. The proposed method appears to be a straightforward application, lacking clarity on any specific challenges or problems it addresses.
- There are no experiments comparing costs. Additionally, it is unclear how the existing models would perform if the size of these models were increased.

**Questions:**

- Tunnel TSP looks like the simplest special form of CTSP. Then can the definition of CTSP easily cover or extend to the one of tunnel TSP too? The comparison between them needs to be specified for better understanding.
- Please provide clear motivations for the target task and proposed approaches. It will help the readers to understand the novelty and importance of this work.
- Is there any challenge when DRL is applied to CTSP?

---

### Official Review · Reviewer_xjCe · 2024-11-03

**Soundness:** 3
**Presentation:** 3
**Contribution:** 2
**Rating:** 3
**Confidence:** 3

**Summary:**

The work aims at a Transformer to solve the Tunnel Traveling Salesmen Problem. Previous single-encoder models are general to distinct vehicle routing tasks but in this work the Transformer is applicable to a specified variant. The performance is incrementally improved since the average optimality gap is still large, and Transformer's applicability obviously weakens.

**Strengths:**

The authors present a comprehensive evaluation of the model, demonstrating its effectiveness across diverse instances of the Tunnel TSP problem.  The proposed approach shows versatility by successfully enhancing multiple neural solvers in addressing the Tunnel TSP, making the design plug-and-play.  The paper is generally well-structured and clearly presented.

**Weaknesses:**

The technical novelty appears limited. The primary contribution centers on introducing a tunnel-specific encoder and corresponding decoding modifications to existing architectures, rather than presenting fundamentally new insights or methodologies.
The method's applicability appears narrowly focused on Tunnel TSP, with insufficient exploration of its potential generalizability to broader combinatorial optimization problems.
The evaluation relies exclusively on synthetic datasets, raising questions about the model's robustness to varying problem sizes and other data distributions.
The computational complexity of the tunnel encoder appears comparable to the node encoder, potentially introducing significant overhead.

**Questions:**

How do the authors incorporate tunnel information for baselines such as POMO?
Could the authors provide a detailed computational analysis, including inference times and parameter counts, to better understand the practical implications of the additional neural network modules?
Given that Tunnel TSP represents a specialized case of Clustered TSP, what are the technical challenges in extending the proposed framework to more general CTSP instances or related vehicle routing problems (e.g., pickup and delivery)?
Could the authors elaborate on concrete real-world applications where their framework provides practical advantages over existing approaches?

---

### Official Review · Reviewer_ZRJH · 2024-11-04

**Soundness:** 3
**Presentation:** 2
**Contribution:** 2
**Rating:** 5
**Confidence:** 2

**Summary:**

The paper addresses the tunnel traveling salesman problem using a deep reinforcement learning approach. It introduces the Double Encoder Transformer (DET) module, which encodes node and tunnel information through two separate encoders. The DET is compatible with existing neural solvers, allowing it to be utilized in a plug and play manner. Experimental results indicate that the proposed DET generally improves the performance of existing neural solvers for tunnel TSP.

**Strengths:**

- Separating node and tunnel information into different encoding pipelines is a reasonable approach to improve the overall performance.
- The plug and play design of DET allows it to integrate smoothly with existing methods, as demonstrated in the experiments.

**Weaknesses:**

- The technical contribution is somewhat limited to the DET module and the separation of the node and tunnel features.
- More explanation on some topics could be beneficial, for example,
    - How are the baseline models trained? Do they explicitly receive tunnel information as inputs to their single-encoders? Or is it only implicitly incorporated through the cost/reward?
    - What is the size of the test samples used to evaluate the models in Table 1? Reporting the variations across multiple training/testing runs would strengthen the claims about DET effectiveness, especially for claims such as guaranteed improvements (Line 468-469).

**Questions:**

Please see Weaknesses.

---

### Official Review · Reviewer_5byA · 2024-11-04

**Soundness:** 2
**Presentation:** 2
**Contribution:** 2
**Rating:** 3
**Confidence:** 4

**Summary:**

This paper tackles a variation of the Traveling Salesman Problem (TSP) referred to as the tunnel TSP, introducing a model called the Double-Encoder Transformer (DET) to solve it. Unlike conventional TSPs, the tunnel TSP includes specific constraints for tunnel traversal, which traditional neural TSP solvers struggle to handle effectively. The proposed DET model enhances existing autoregressive neural TSP solvers by incorporating separate encoders for nodes and tunnels, allowing the model to more accurately process the unique interactions between these elements in the tunnel TSP. The authors demonstrate that integrating DET into established neural solvers (such as POMO) can reduce the optimality gap for tunnel TSP, enhancing solution quality.

**Strengths:**

The DET model's architecture, with separate encoders for nodes and tunnels, is a practical adaptation that enables better handling of tunnel-specific constraints within TSP solutions. The authors demonstrate measurable performance improvements over existing solvers for this problem variant, validating DET’s utility in improving optimality gaps in tunnel TSP instances.

**Weaknesses:**

While DET shows practical utility, its novelty is limited due to its reliance on well-established architectures (POMO) and scoring techniques (regret). The approach primarily focuses on adapting feature encoding without introducing significant new concepts in neural TSP solving or reinforcement learning. Moreover, the evaluation lacks comparisons with a broader range of TSP variants and solvers, which would better contextualize DET’s relative efficacy. Lastly, the choice of DET may result in increased computational overhead due to the dual encoder, which the paper does not address in terms of efficiency or resource requirements.

**Questions:**

How does the proposed DET model perform in other TSP variations or combinatorial optimization tasks that have similar clustering constraints?

Could further feature augmentation, beyond tunnel information, bring improvements, or would such modifications saturate the model's performance gains?

---

### Note · Authors · 2024-11-15

I have read and agree with the venue's withdrawal policy on behalf of myself and my co-authors.